# One-Dimensional Photonic Crystal with a Defect Layer Utilized as an Optical Filter in Narrow Linewidth LED-Based Sources

Michal Gryga *, Dalibor Ciprian , Lucie Gembalova and Petr Hlubina *

Department of Physics, Technical University Ostrava, 17. Listopadu 2172/15, 708 00 Ostrava-Poruba, Czech Republic

* Correspondence: michal.gryga@vsb.cz (M.G.); petr.hlubina@vsb.cz (P.H.)

**Abstract:** A one-dimensional photonic crystal (1DPhC) with a defect layer is utilized as an optical filter in a simple realization of narrow linewidth LED-based sources. The 1DPhC comprising $TiO_2$ and $SiO_2$ layers is characterized by two narrow defect mode resonances within the 1DPhC band gap, or equivalently, by two peaks in the normal incidence transmittance spectrum at wavelengths of 625.4 nm and 697.7 nm, respectively. By combining the optical filter with LEDs, the optical sources are employed in interferometry experiments, and the defect mode resonances of a Lorentzian profile with linewidths of 1.72 nm and 1.29 nm, respectively, are resolved. In addition, a simple way to tune the resonances by changing the angle of incidence of light on the optical filter is demonstrated. All-dielectric optical filters based on 1DPhCs with a defect layer and combined with LEDs thus represent an effective alternative to standard coherent sources, with advantages including narrow spectral linewidths and variable output power, with an extension to tunable sources.

**Keywords:** one-dimensional photonic crystal; defect layer; defect mode; transmittance; filter; Lorentzian profile; linewidth



## 1. Introduction

One-dimensional photonic crystals (1DPhCs) or distributed Bragg reflectors (DBRs) as alternatives to complex dielectric structures [1] with a periodic modulation in the refractive indices have attracted enormous interest in recent years. Due to alternating high and low refractive indices within the layers of the structure and wave interference, the so-called photonic band gap (PBG) is present for which light propagation is forbidden. When compatible defects are introduced in the structure, defect modes appear within the PBG, and the 1DPhCs thus represent an electromagnetic counterpart of semiconductor crystals with many promising applications. These include optical filters [2–6], along with tunable ones [7–12], lasers [13], light-emitting diodes [14], active cavities [15], including optical field enhanced nonlinear absorption [16], optical sensors [17–19], fiber-optic sensors [20], humidity sensors [21], gas sensors [22–25], and photonic biosensors [26–32]. In addition, alternative designs to the 1DPhC-based optical filters are represented by the 2DPhC-based ones [33–35].

Among optical sensors, Bloch-surface-wave (BSW)-based sensors [36–49] are the most promising. This is owed to the tunability of 1DPhCs to support BSWs in any desired wavelength range by varying the dispersion and geometry of the 1DPhC. Moreover, BSWs can be excited by both *s*- and *p*-polarized waves [47,48] at any wavelength, including the Vis and NIR regions, which represents one of the advantages compared to surface plasmon polaritons (SPPs) [44,49], which can be exited by a *p*-polarized wave only. Among other advantages are sharper resonances due to absence of a metal layer, high field enhancement leading to high sensitivity and, thus, a higher figure of merit, and longer propagation distances [50]. However, to excite BSWs, similar to SPPs, some coupling element (usually a prism) needs to be employed to fulfill the phase-matching condition for the surface wave, in which strong confinement of light is attained.

Because a direct free-space excitation of both BSWs and SPPs is not possible, suitable alternatives such as Tamm plasmons (TPs) [51–59] or defect modes [2–5,7–32] are used. TPs represent waves at the interface between a metal and a 1DPhC, and this concept has a number of sensing applications [55–59]. Defect modes are related to all-dielectric 1DPhCs with defect layers or optical cavities [2–5,7–11,13–31]. A defect layer introduced in the 1DPhC permits the existence of defect states that are characterized by strong confinement of light in the resonant cavity. Moreover, the strong confinement of light is manifested by very narrow resonances within the PBG, even at the normal incidence of light, both in transmission and reflection. The position of the defect state in the PBG, or equivalently, the resonance wavelength, is given by the refractive index and thickness of the defect layer. A high refractive index leads to a defect mode with the maximum intensity, while the narrower resonance in the defect mode is due to increasing the number of layers [60].

1DPhCs with cavity mode resonances represent an effective alternative to various applications, as experimentally confirmed [2–5,7–11,13–24]. One of them refers to narrow-band transmission filters and their potential applications in narrow linewidth sources. Even if the maximum transmission of the filter is limited, high-power sources such as LEDs are available, and realizations of narrow linewidth sources of a sufficient power are possible. Moreover, because all-dielectric 1DPhCs are characterized by low losses, the absorption of light is negligible. This concept, to the best of the authors' knowledge, represents the first demonstration of narrow linewidth LED-based sources that can be utilized in a wide range of applications.

In this paper, a 1DPhC with a defect layer exhibiting resonances within the 1DPhC band gap is analyzed theoretically and experimentally. The 1DPhC represents an optical filter and is utilized in a simple realization of narrow linewidth LED-based sources. For the 1DPhC comprising $TiO_2$ and $SiO_2$ layers, two peaks are revealed in the normal incidence transmission spectrum. The filter combined with LED sources led to the defect mode resonances of a Lorentzian profile [15], and an interferometric method resolved linewidths of 1.72 nm and 1.29 nm at wavelengths 625.4 nm and 697.7 nm, respectively. In addition, we revealed that, by changing the angle of incidence of light on the optical filter, the resonance wavelengths can be changed so that a tunable source of variable power can be realized. All-dielectric optical filters based on defect mode resonances in 1DPhCs and combined with LEDs thus represent an effective alternative to available coherent sources, with advantages including narrow spectral linewidths and variable output power.

## 2. Material Characterization

The multilayer structure under consideration, which represents a 1DPhC with a defect layer (an optical cavity with distributed Bragg reflectors), is characterized by a normal-incidence band gap in the reflection spectrum approximately 200 nm wide (580–780 nm). The 1DPhC was fabricated on a glass substrate by a thin film sputtering deposition technique, and the layer thicknesses of $TiO_2$ and $SiO_2$ were provided by the manufacturer (Meopta, Czech Republic) and estimated from an image obtained by the technique of scanning electron microscopy (SEM). From an SEM image (see Figure 1) captured by a scanning electron microscope (Quanta 650 FEG, USA), it was confirmed that the 1DPhC is composed of a system of twenty-one layers of $TiO_2$ and $SiO_2$, including the $SiO_2$ defect layer (optical cavity) with the largest thickness. We found that the layers of $TiO_2$ ($i = 1, \ldots, 11$) have thicknesses $t_i$, and similarly, the layers of $SiO_2$ ($j = 1, \ldots, 10$) have thicknesses $t'_j$, as schematically shown in Figure 2; their values are specified in Table 1.

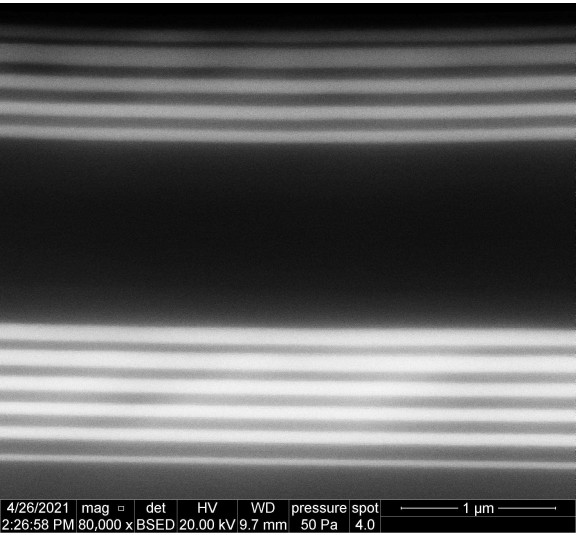

**Figure 1.** An SEM image of a 1DPhC comprising TiO$_2$ (white) and SiO$_2$ layers.

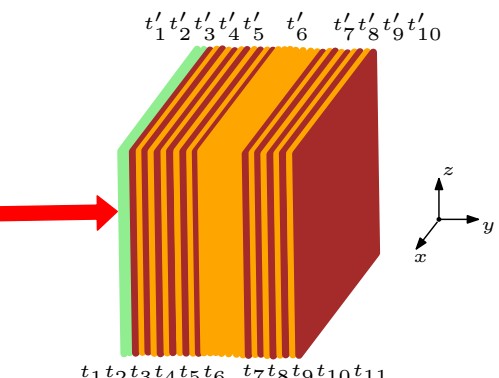

**Figure 2.** A 1DPhC with a defect layer comprising layers of TiO$_2$ (dark color) and SiO$_2$ (light color).

**Table 1.** TiO$_2$ and SiO$_2$ layer thicknesses.

| TiO$_2$ Layer | Thickness (nm) | SiO$_2$ Layer | Thickness (nm) |
|---|---|---|---|
| $t_1$ | 54.6 | $t'_1$ | 127.5 |
| $t_2$ | 76.5 | $t'_2$ | 80.0 |
| $t_2$ | 94.8 | $t'_3$ | 80.0 |
| $t_4$ | 102.0 | $t'_4$ | 80.0 |
| $t_5$ | 105.9 | $t'_5$ | 76.5 |
| $t_6$ | 113.4 | $t'_6$ | 1206.0 |
| $t_7$ | 91.1 | $t'_7$ | 90.0 |
| $t_8$ | 94.8 | $t'_8$ | 100.0 |
| $t_9$ | 94.8 | $t'_9$ | 100.0 |
| $t_{10}$ | 127.5 | $t'_{10}$ | 50.0 |
| $t_{11}$ | 67.4 | | |

The refractive index dispersion of a glass substrate and the layers of SiO$_2$ and TiO$_2$ was obtained from the ellipsometric data for a 1DPhC including the same layer materials [61]. In the case of the glass substrate, the refractive index (RI) as a function of wavelength is expressed by the Cauchy formula:

$$n_{sub}(\lambda) = A - B\lambda + C\lambda^2 - D\lambda^3, \tag{1}$$

where the values of the constants are $A = 1.51824$, $B = 0.19112 \, \mu\text{m}^{-1}$, $C = 0.019391 \, \mu\text{m}^{-2}$, and $D = 0.07108 \, \mu\text{m}^{-3}$ for wavelength $\lambda$ in micrometers. The RI dispersion of thin films is described by the formula:

$$n_i^2(\lambda) = A + \frac{B\lambda^2}{\lambda^2 - C^2} - D\lambda^2, \tag{2}$$

where index $i$ = TiO$_2$, SiO$_2$ indicates the material and constants $A, B, C$, and $D$ for TiO$_2$ are $A = 0$, $B = 4.672$, $C = 0.22935 \, \mu\text{m}$, and $D = 0 \, \mu\text{m}^{-2}$ and for SiO$_2$ are $A = 1.348$, $B = 0.756$, $C = 0.10683 \, \mu\text{m}$, and $D = 0.00975 \, \mu\text{m}^{-2}$, respectively.

### 3. Theoretical Analysis

*Spectral Transmittance*

To analyze the response of the multilayer structure under study, representing a 1DPhC with a defect layer, the spectral transmittance $T(\lambda)$ can be evaluated using the transfer matrix method (TMM). If $N$ dielectric layers are considered, the transmission matrices across different interfaces and propagation matrices in different homogeneous dielectric media [46] need to be evaluated to obtain the total transfer matrix $\mathbf{M}(\lambda)$ at the wavelength $\lambda$:

$$\mathbf{M}(\lambda) = \begin{bmatrix} M_{11}(\lambda) & M_{12}(\lambda) \\ M_{21}(\lambda) & M_{22}(\lambda) \end{bmatrix} = \tag{3}$$

$$= \left[ \prod_{j=1}^{N} \mathbf{B}_{j-1,j}(\lambda)\mathbf{P}_j(\lambda) \right] \cdot \mathbf{B}_{N,N+1}(\lambda), \tag{4}$$

where index 0 refers to the first semi-infinite medium and $N + 1$ to the last one. Considering the normal incidence conditions, the boundary matrices $\mathbf{B}_{j,j+1}(\lambda)$ are expressed as

$$\mathbf{B}_{j,j+1}(\lambda) = \frac{1}{2} \begin{pmatrix} 1 + \eta(\lambda) & 1 - \eta(\lambda) \\ 1 - \eta(\lambda) & 1 + \eta(\lambda) \end{pmatrix}, \tag{5}$$

with the parameter $\eta(\lambda)$ given as

$$\eta(\lambda) = \frac{n_{j+1}(\lambda)}{n_j(\lambda)}, \tag{6}$$

where $n_j(\lambda)$ and $n_{j+1}(\lambda)$ are the media refractive indices. The propagation matrices are expressed as

$$\mathbf{P}_j(\lambda) = \begin{pmatrix} e^{i\frac{2\pi}{\lambda}n_j(\lambda)t_j} & 0 \\ 0 & e^{-i\frac{2\pi}{\lambda}n_j(\lambda)t_j} \end{pmatrix}, \tag{7}$$

where $t_j$ is the thickness of the $j$-th layer.

The spectral transmittance $T(\lambda)$ is calculated using the total transfer matrix element as

$$T(\lambda) = \frac{1}{|M_{11}(\lambda)|^2}. \tag{8}$$

To model the transmittance spectrum $T(\lambda)$ for the 1DPhC with a defect layer shown in Figure 2, we considered air as the surrounding medium and the incident light entering the glass substrate.

The thicknesses and dispersion of the materials of the 1DPhC specified above were taken into account, and the extinction coefficients for the TiO$_2$ and SiO$_2$ layers, $\kappa_{\text{TiO}_2} = 1.6 \times 10^{-3}$ and $\kappa_{\text{SiO}_2} = 3.4 \times 10^{-4}$ [46], respectively, were assumed. The theoretical spectral transmittance shown in Figure 3a demonstrates two narrow peaks at wavelengths of 635.5 nm and 715.7, respectively, which manifest the excitation of the defect modes. The short-wavelength peak is due to the Bragg mirrors, while the long-wavelength one

corresponds to the cavity of the considered structure (with one resonance peak only). The long-wavelength peak is shown in detail together with a Lorentzian function in the wavelength domain, which results from its angular frequency $\omega$ representation [15,62]:

$$T(\omega) = T(\omega_0)\frac{\Gamma^2}{(\omega - \omega_0)^2 + \Gamma^2},$$

(9)

where $\omega_0$ is the central angular frequency, $T(\omega_0)$ is the corresponding transmittance, and $\Gamma$ is the spectral half width. The conversion from the angular frequency domain to the wavelength domain can be performed using $\omega = 2\pi c/\lambda$ and $\Gamma/\omega = -\Delta\lambda_{1/2}/\lambda$, and Figure 3b illustrates the excellent approximation of the resonance peak by the Lorentzian function with the parameters: the central wavelength $\lambda_0 = 715.74$ nm, the half-width $\Delta\lambda_{1/2} = 1$ nm, and the maximum transmittance $T(\lambda_0) = 0.52$.

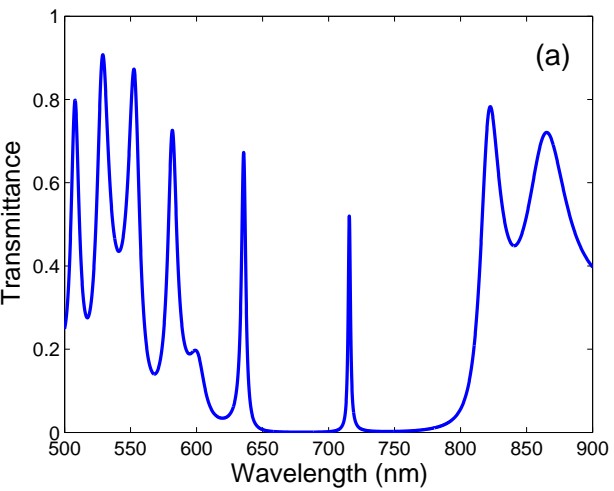 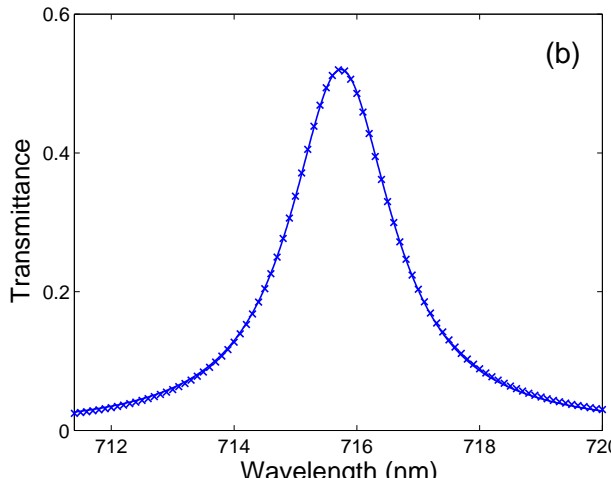

**Figure 3.** Theoretical spectral transmittance $T(\lambda)$ at normal incidence for the 1DPhC (**a**). The long-wavelength peak (crosses) fit to the Lorentzian spectrum (solid curve) (**b**).

To confirm the defect mode excitation, the normalized optical field intensity $|E_x|^2/|E_{x0}|^2$ in the structure at a wavelength of 715.74 nm is shown in Figure 4a. The computation was performed using the TMM [63] when the coordinate system shown in Figure 2 was used, and $E_{x0}$ represents the $x$ component of the electric field of the incident wave. From the figure, the optical field enhancement in the defect layer is apparent, with more than a 39-fold enhancement of the optical intensity with respect to the incident beam. Moreover, this is accompanied by the formation of standing waves in the defect layer (cavity) due to the constructive interference of the incident and reflected waves, resulting in a very narrow peak in the transmission spectrum.

A simple way to tune the defect mode resonances is changing the angle of incidence of light on the 1DPhC, as shown in Figure 4b for the angle of incidence $\alpha$ increased up to 25°. It is clearly seen that the resonances are shifted and differ in the amplitude and also slightly in the linewidth.

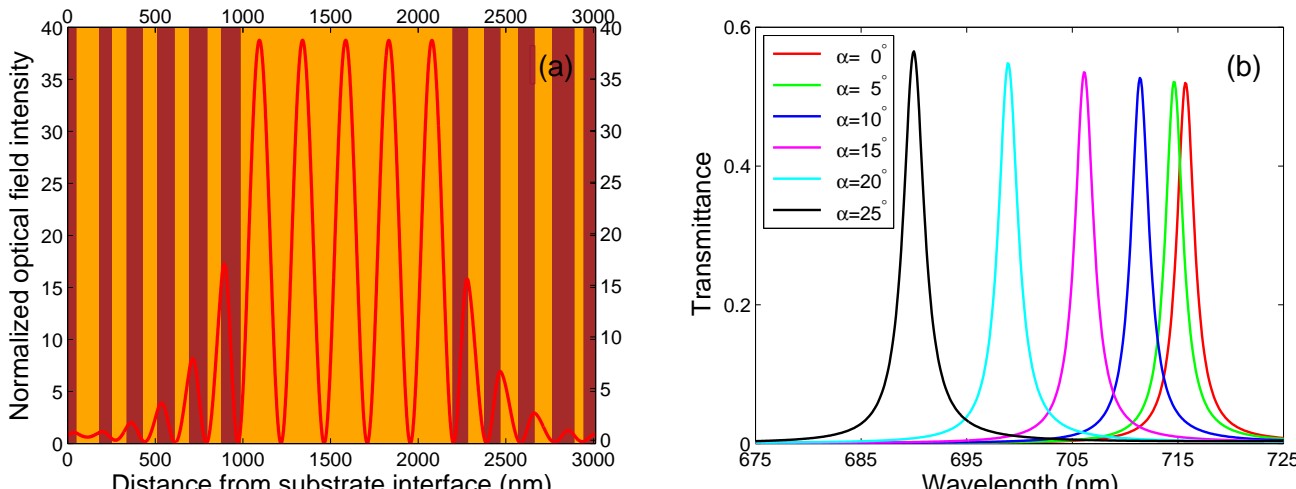

**Figure 4.** The normalized optical field intensity distribution for the optical wave at a wavelength of 715.74 nm (**a**). The spectral transmittance $T(\lambda)$ for various angles of incidence $\alpha$ (**b**).

## 4. Experimental Setups

At first, the spectral transmittance $T(\lambda)$ of a 1DPhC with a defect layer was measured using the setup shown in Figure 5, where a white light source (WLS) was used. Light from the WLS (halogen lamp HL-2000, Ocean Optics, Dunedin, FL, USA) is launched into an input optical fiber (IOF) terminated by a collimation lens (CL) to generate a collimated beam of light incident at angle $\alpha$ on the 1DPhC. The transmitted light is launched via a microscope objective (MO) into a read optical fiber (ROF) of a compact fiber-optic spectrometer (USB4000, Ocean Optics) to measure the transmittance spectrum in a wavelength range of 400–1000 nm. The transmission spectrum was normalized with respect to the signal recorded by the spectrometer without the 1DPhC.

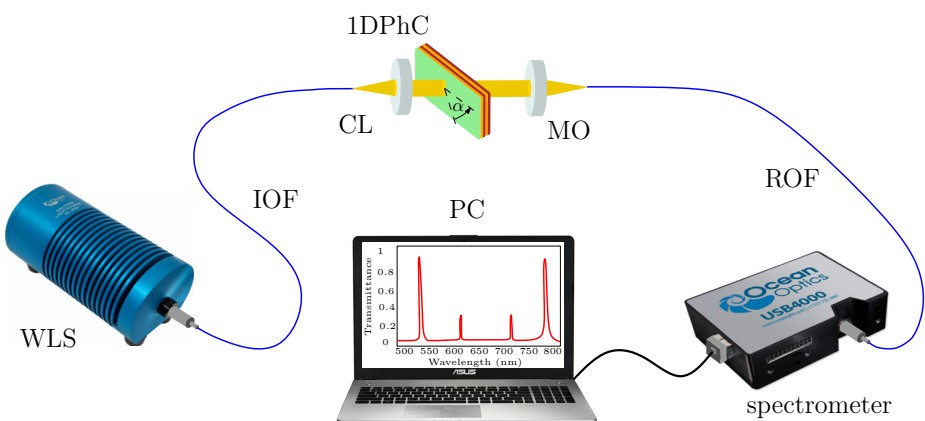

**Figure 5.** Experimental setup for measuring the spectral transmittance $T(\lambda)$ of a 1DPhC; white light source (WLS), input optical fiber (IOF), collimation lens (CL), one-dimensional photonic crystal (1DPhC), microscope objective (MO), read optical fiber (ROF), personal computer (PC).

Because narrow spectral resonance peaks are lowered and broadened due to the limiting resolving power of the spectrometer [19], the method of Fourier transform interferometry was applied to overcome this limitation. The method is based on the measurement of the amplitude of a coherence function, or equivalently, a visibility function [62] from which the linewidth (full-width at half-maximum) $\Delta\lambda$ can be obtained. The setup used to measure the visibility function is shown in Figure 6, and it comprises a Michelson interferometer with an LED and a defective 1DPhC at its input and a monochrome USB 3.0 camera (Basler, acA2440-75 μm) to record a spatial interferogram. Light from an LED is

launched into an input optical fiber (IOF) terminated by a collimation lens (CL) to generate a collimated beam of light. The beam excites the Michelson interferometer comprising the beam splitter (BS) and Mirrors 1 and 2 (M1 and M2), with one of them moveable to adjust the path difference. At the output, a camera is used to capture an interferogram by a personal computer (PC).

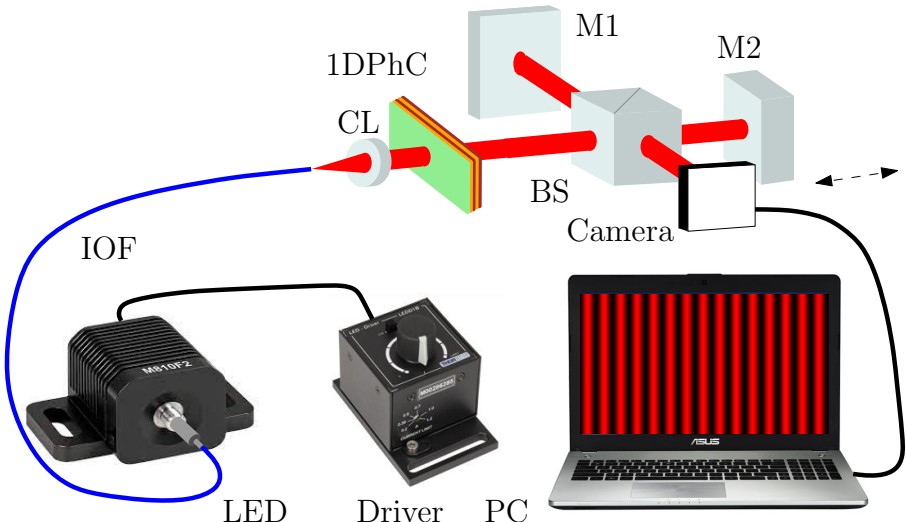

**Figure 6.** Experimental setup for measuring a visibility function of the interference field in a Michelson interferometer excited by an LED combined with a 1DPhC; light-emitting diode (LED), input optical fiber (IOF), collimation lens (CL), one-dimensional photonic crystal (1DPhC), beam splitter (BS), Mirror 1 (M1), Mirror 2 (M2), personal computer (PC).

## 5. Experimental Results and Discussion

First, the experimental setup shown in Figure 5 was used to measure the spectral transmittance $T(\lambda)$ for the 1DPhC with the defect layer at the normal incidence of light when the surrounding medium was air. The measured spectral transmittance $T(\lambda)$ is depicted in Figure 7a and illustrates, in accordance with the theory, that the cavity mode excitations show up as narrow peaks within the band gap, which is approximately 210 nm wide (580–790 nm), with the amplitude greater for the short-wavelength peak.

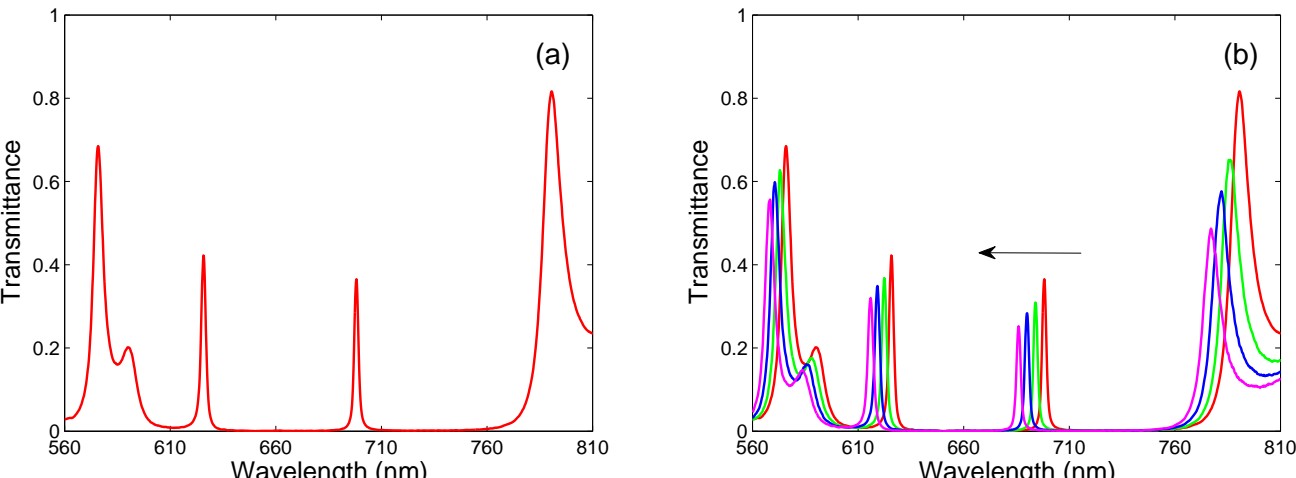

**Figure 7.** Spectral transmittance $T(\lambda)$ measured at the normal incidence (**a**) and for increasing angle of incidence given by arrow (**b**).

In Figure 7b, the spectral transmittance $T(\lambda)$ is shown when the measurement was extended to the oblique incidence of light on the 1DPhC. It is clearly seen that both peaks

shift, in accordance with the theory, towards short wavelengths, but their amplitude, contrary to the theory, decreases with increasing angle of incidence. The decrease can be attributed to the out-of-axis detection of the output collimated beam, which is shifted due to the refraction of light in the 1DPhC. Moreover, the shift of the resonance peaks is adjusted nearly constant due to the rotation of the 1DPhC with variable angle increments.

Next, the narrow resonances were utilized in narrow-band filters employed in narrow linewidth LED-based sources. To illustrate the operation of such sources, we utilized two commercially available LEDs of central wavelengths 625 nm (M625F2, Thorlabs, Newton, MA, USA) and 700 nm (EP700S04, Thorlabs), respectively. The filtering of the LED spectrum employing the 1DPhC with the defect mode, similar to as demonstrated in a previous paper [19], is illustrated in Figure 8a,b. First, the LED spectrum is filtered to a narrow spectrum with the maximum transmittance $T(\lambda_0) = 0.39$, as shown in Figure 8a. The transmission spectrum $T(\lambda)$ is fit to a Lorentzian function according to Equation (9) with the central wavelength $\lambda_0 = 625.4$ nm and the linewidth $\Delta\lambda = 2.42$ nm. Similarly, the same procedure is applied to the second LED spectrum, which is filtered to a narrow spectrum with the maximum transmittance $T(\lambda_0) = 0.34$, as shown in Figure 8b, with the central wavelength $\lambda_0 = 697.7$ nm and the linewidth $\Delta\lambda = 2.02$ nm.

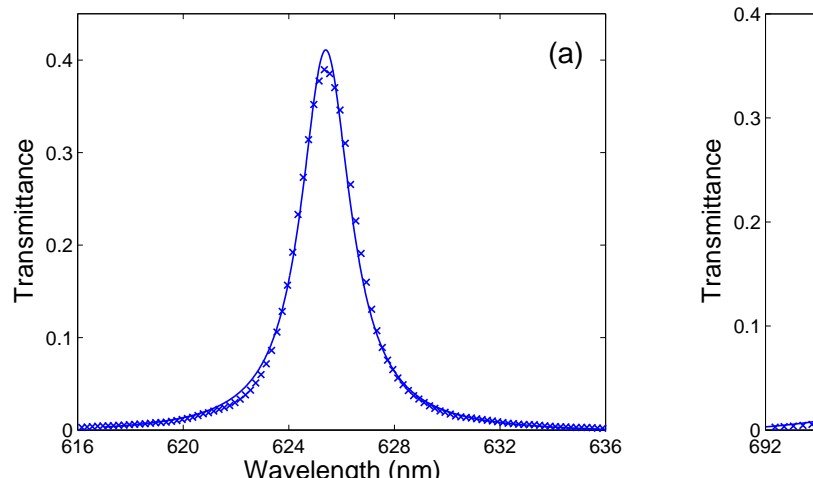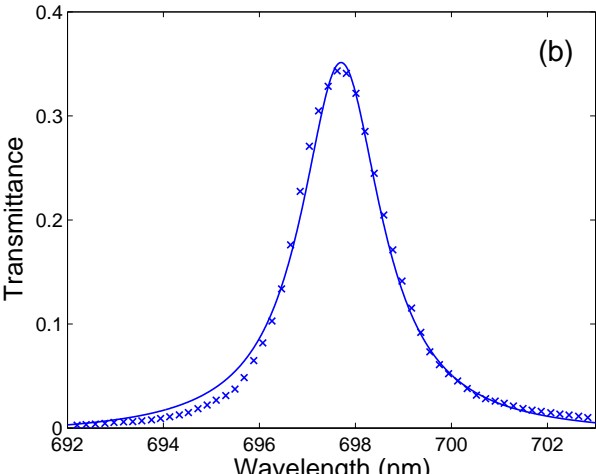

**Figure 8.** Measured transmittance $T(\lambda)$ for the 1DPhC and two LED sources: M625F2 (**a**) and EP700S04 (**b**).

However, the linewidth $\Delta\lambda$ obtained by the processing of the measured transmission spectrum $T(\lambda)$ using a Lorentzian function is not the actual one. To measure the actual linewidth, the method of Fourier transform interferometry was applied. A coherence function, which corresponds to the Lorentzian spectral line, gives a visibility function for the spatial interference fringes according to [62]:

$$V(\Delta l) = V_0 \exp(-|\Delta l|/l_c), \tag{10}$$

where $V_0$ is the maximum visibility, $\Delta l$ is the path difference in the interferometer, and $l_c$ is the coherence length.

To measure the visibility function $V(\Delta l)$ for the first LED-based source, a Michelson interferometer shown in Figure 6 was used and the interferogram captured by a camera was processed for different path lengths $\Delta l$. The processing of the interferogram includes smoothing and evaluation of the intensity, as illustrated in a previous paper [64], to obtain their extremal values and express the visibility. As an example, an interferogram composed of the spatial interference fringes with a visibility of 0.606 is shown in Figure 9a for the smallest path difference. Figure 9b then shows the measured visibility function together with the theoretical one corresponding to Equation (10) with the coherence length $l_c = 72.5$ μm and the linewidth $\Delta\lambda = 1.72$ nm, which is smaller than the one obtained

from the spectral measurement shown in Figure 8a. The highest visibility is lower than one owing to the effects of the excitation part of the setup (the input multimode optical fiber of a given core diameter and the collimation lens), justifying the difference from the point-like source.

A similar procedure was used for the second LED-based source, and an example of the interferogram for the smallest path difference, composed of the spatial interference fringes with a visibility of 0.545, is shown in Figure 10a. Next, the measured visibility function is shown in Figure 10b together with the theoretical one, which gives the coherence length $l_c = 121.1$ μm. The corresponding linewidth $\Delta\lambda = 1.29$ nm is smaller than that for the first LED-based source. To obtain narrower band and highly transmissive optical filters, a larger number of alternating dielectric layers needs to be considered, including, for example, a photonic heterostructure [65]. Moreover, the environmental perturbations such as changes in temperature [24,66] and humidity can affect the filter central wavelength. Extending the considerations to special defect layers such as porous [58], gaseous [24], and liquid [67] ones, various optical sensors can be proposed.

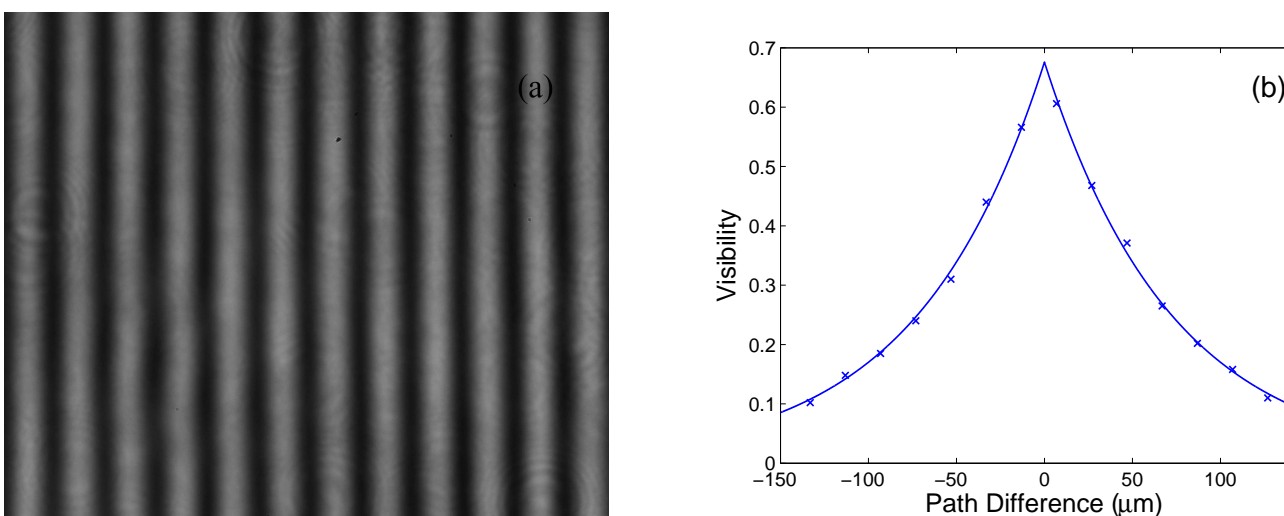

**Figure 9.** Interferogram captured by a camera for the first LED-based source and the smallest path difference (**a**). Measured visibility function (crosses) with a fit (**b**).

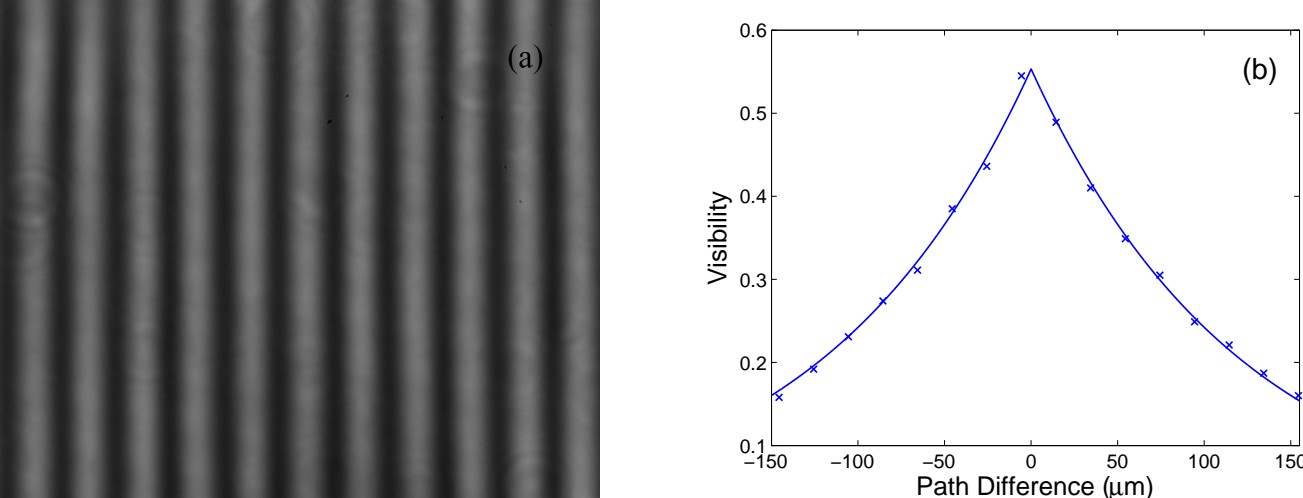

**Figure 10.** Interferogram captured by a camera for the second LED-based source and the smallest path difference (**a**). Measured visibility function (crosses) with a fit (**b**).

Previous results with a white light source (see Figure 7b) indicated that the spectra of the LED-based sources can be shifted by changing the angle of incidence of light on the 1DPhC, and thus, simple tunable sources can be realized. This is illustrated for the first LED-based source in Figure 11a showing the transmitted power for increasing angle of incidence. The spectrum shifts towards short wavelengths and decreases due to the spectrum of the LED source [19]. The LED-based source of a sufficient output power has a tunability range approximately from 612 nm up to 625.4 nm, with 39% transmission of the LED signal at 625.4 nm. Similarly, the output spectral power of the second LED-based source for increasing angle of incidence is shown Figure 11b. The tunability range of the source with a sufficient output power is wider, and it is approximately from 672 up to 697.7 nm, with 34% transmission of the LED signal at 697.7 nm.

Generally, the central wavelength and output power of the narrow-linewidth LED-based sources can be tuned by a fine adjustment of the angle of incidence of a collimated light beam on the 1DPhC. In addition, the collimated beam represents a substantial advantage compared to a converging beam generating a spot of sub-millimeter diameter on the surface of a 1DPhC [68], especially for a filter employing an open cavity between two DBRs [19].

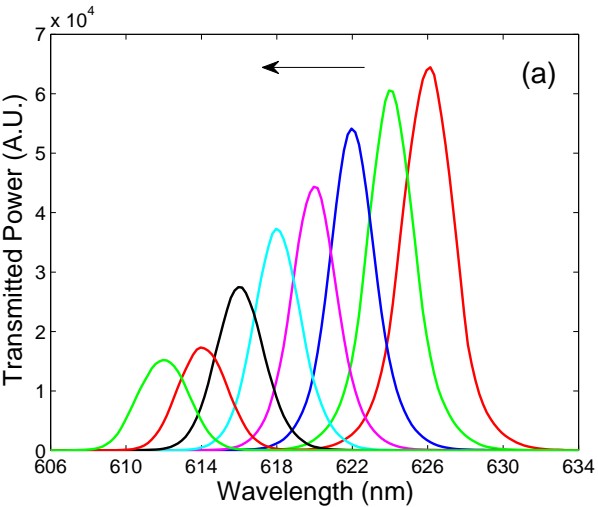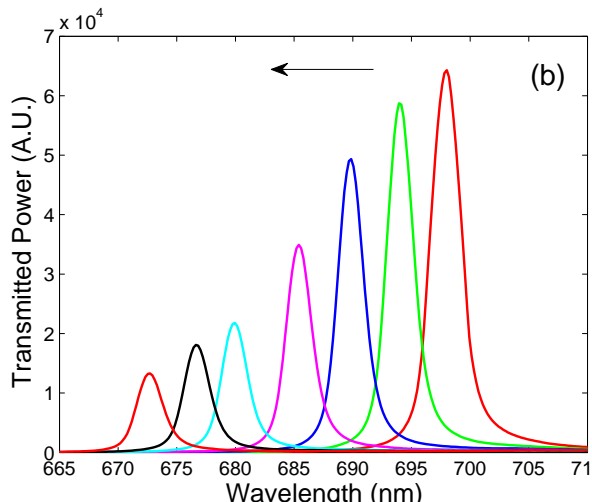

**Figure 11.** Measured spectral power transmission of the LED-based sources for increasing angle of incidence given by arrow: the first LED (**a**) and the second LED (**b**).

From various designs of 1DPhCs that have been used for optical filters [2–12] working in the NIR [2,3,5,8,10], UV [4], and Vis [6,7,9,11,12] spectral regions, only a few have been realized [4–9,11]. Our spectral filter outperforms them in the linewidth.

## 6. Conclusions

In this paper, a 1DPhC with a defect layer exhibiting resonances within the 1DPhC band gap was analyzed theoretically and experimentally. The transmittance spectra of the 1DPhC comprising layers of $TiO_2$ and $SiO_2$ were modeled at the normal and oblique incidence of light, and narrow peaks were resolved. In addition, a Lorentzian shape of a resonance peak was revealed.

The theoretical results were confirmed by the experimental ones. We showed that the 1DPhC with the defect layer of a suitable transmittance spectrum can be employed as a spectral filter, and this was demonstrated for two LED-based sources at wavelengths of 625.4 nm and 697.7 nm, respectively. By combining the optical filter with the LED sources, the defect mode resonances of a Lorentzian profile were resolved, and their linewidths of 1.72 nm and 1.29 nm, respectively, were measured by an interferometric method.

All-dielectric optical filters based on 1DPhCs with a defect layer and combined with LEDs thus represent an effective alternative to available coherent sources, with advantages

including narrow spectral linewidths and variable output power, with an extension to tunable sources. The concept can be extended to open cavities between the distributed Bragg reflectors with a number of applications in the field of optical sensors.

**Author Contributions:** Conceptualization, P.H.; methodology, M.G., D.C. and P.H.; software, M.G., D.C. and P.H.; validation, M.G., D.C. and P.H.; formal analysis, M.G., D.C. and P.H.; investigation, M.G., D.C., L.G. and P.H.; resources, P.H.; data curation, M.G., D.C. and P.H.; writing—original draft preparation, M.G. and P.H.; writing—review and editing, M.G., D.C., L.G. and P.H.; visualization, M.G., D.C., L.G. and P.H.; supervision, P.H.; funding acquisition, P.H. All authors have read and agreed to the published version of the manuscript.

**Funding:** The research was supported by the student grant system through Project SP2023/046 and by ERDF/ESF Project No. CZ.02.1.01/0.0/0.0/17_048/0007399.

**Institutional Review Board Statement:** Not applicable.

**Informed Consent Statement:** Not applicable.

**Data Availability Statement:** Data underlying the results presented in this paper are not publicly available at this time, but may be obtained from the authors upon reasonable request.

**Conflicts of Interest:** The authors declare no conflict of interest.

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
