# Peer review of "One-Dimensional Photonic Crystal with a Defect Layer Utilized as an Optical Filter in Narrow Linewidth LED-Based Sources"

_crystals, doi:10.3390/cryst13010093_

Round 1
Reviewer 1 Report
In the submitted manuscript authors have presented a optical filter to realize a narrow linewidth LED-based sources using one-dimensional photonic crystal (1DPhC) with a defect layer. The presented 1D-PhC cavity structure concept is well know and well explored in the literature but the considered methodology is unique. The paper may be accepted after addressing following queires:
1. In abstract authors mentioned "narrow linewidth LED-based sources" Here, the Narrow bandwidth is used for LED or filter have narrow bandwidth?
2. The obtained line width of 1.72nm is very high, can it further be improved?
3. Fig. 2: mention the materials with the color.
4. What could be the possible reason to obtain two resonance peaks, because for the considered structure there must be one resonance peak only.
5. How the environmental perturbation (temperature, humidity etc) will impact the working of device. There are many recent work on TiO2 and SiO2 based materials, who reported the temperature dependent performance variation of device. Authors should comment on this.
Reviewer 2 Report
This paper uses a one-dimensional photonic crystal with a defect layer as an optical filter. The chosen topic is interesting for the readers, and the structure of the article and its text are appropriate to the article's subject. This structure consists of TiO2 and SiO2 layers. The PBG of the structure is 200 nm and is in the range of visible light. There are concerns with this article that need to be addressed before publication.
1. By definition, a photonic crystal is a structure that has periodicity. This means that the thickness of the layers must be the same. But in the designed structure, the thickness of homogeneous layers (for example, SiO2) is not the same. Is this discrepancy due to a manufacturing error, or was the author's intention?
2. Various designs have been made using a one-dimensional photonic crystal for the filter. The comparison of the results to these structures and the advantage of the proposed work should be stated at the end of the article,
3. Two-dimensional photonic crystals are structures that can confine light in more directions. It is better to explain two-dimensional photonic crystal structures and the reason for choosing a one-dimensional one. For example, references can be added, and an explanation of two-dimensional structures can be added to the text:
https://doi.org/10.1016/j.optcom.2018.07.039, https://doi.org/10.1007/s11082-022-03945-9, https://dx.doi.org/10.3923/jas.2008.1891.1897
4. What is the unit of vertical graphs in figures 7-11?
5. This design's innovation and outstanding features compared to previous works can be added in the abstract section.
Round 2
Reviewer 2 Report
In the new version of the manuscript, all comments have been answered. Therefore, the article is suitable for publication in the journal.